# Value Stream Mapping as a Supporting Management Tool to Identify the Flow of Industrial Waste: A Case Study

**Yolandi Schoeman** [1,*] [ID] **, Paul Oberholster** [1] **and Vernon Somerset** [2]

1   Centre for Environmental Management, Faculty of Natural and Agricultural Sciences,
    University of the Free State, Bloemfontein 9301, South Africa; oberholsterpj@ufs.ac.za
2   Chemistry Department, Faculty of Applied Sciences, Cape Peninsula University of Technology, Bellville 7535,
    South Africa; somersetv@cput.ac.za
*   Correspondence: schoeman.yolandy@gmail.com

**Abstract:** The Value Stream Mapping (VSM) method was applied to a case study in the iron and steel industry in Southern Africa as a supporting management tool to identify, demonstrate, and evaluate industrial waste and comprised of three steps. The first step included collecting and verifying waste generation and flow data as the VSM data input step. The second step comprises three phases: mapping waste generation and fractions and horizontal and vertical performance analysis. The third step is comprised of actual and future state maps compilation. Following the first year of implementation, waste was reduced by 28%, and waste removal cost by 45%. Implementing the VSM method demonstrated cost savings and reduced waste flow within the study's first year. The initial waste generation reduction target of 5% per annum was exceeded. The VSM method application proved to be a practical method for the iron and steel industry to visualize and analyze waste flows, identify opportunities and challenges in waste management operations, reduce waste, promote lean manufacturing, and achieve an environmentally responsible zero-waste environment.

**Keywords:** value stream mapping; industrial waste; lean manufacturing; zero-waste; waste management

## 1. Introduction

Iron and steel manufacturing is one of the most fundamental industrial processes globally [1] and accounts for a vital sector to national economies [2]. Sustainability in the iron and steel industry remains a goal for the modern society and includes sustainable iron and steelmaking goals as (i) conserving natural resources, (ii) reducing greenhouse gas emissions, (iii) reducing volatile emissions, (iv) reducing landfill waste and (v) reducing hazardous waste [1,3]. Reducing landfill waste and hazardous waste remains a complex issue that iron and steel companies globally must address daily [4]. Waste generation in the iron and steel industry translates to inefficacy and ineffectiveness in manufacturing performance [2]. Generating waste is associated with negative consequences such as environmental degradation, social impacts, and economic expenses to mitigate environmental impacts that further challenge the sustainability of the iron and steel industry.

There has been an increased focus on reducing material waste [5] in manufacturing, and especially in the iron and steel industry, to promote the movement towards sustainability [6] and to implement lean methods and tools as part of operational activities [7]. Other concepts introduced to promote sustainability includes implementing environmental management standards such as ISO 14001, natural capitalism, Factor 10, ecological footprinting, the natural step framework, and other sustainable management visions, norms, and directions [6]. Methods and tools available for environmental management [8] include material flow cost accounting [9,10] and cleaner production approaches [11], and without prescribed methods in specific standards, such as ISO 14000, companies use different tools [6]. Additionally, capturing manufacturing value can be enabled by identifying

innovative resource-efficient business models and solutions [12]. The latter can be combined with technologies, manufacturing methods [13], and decision support systems to aid industrial waste management [14].

Limitations exist in mapping iron and steel waste generation and fractions and analyzing waste systems' performance to demonstrate material efficiency in waste segments in iron and steel industries in developing countries in Southern Africa. A solution to this shortcoming is to apply Value Stream Mapping (VSM) in manufacturing to promote lean manufacturing and sustainability [15–19]. Applying VSM includes completing the process of illustrating, identifying, and industrial waste measuring resulting in compiling industrial waste flow mapping and diagrams for "actual" and "future" waste flow mapping [7]. The principal aim of VSM is to identify industrial waste in manufacturing systems [16], which can have a positive impact on environmental performance [7]. Ref. [20] Agrees that VSM magnifies the benefits of environmental performance through less energy and waste. Ref. [21] Found that the VSM can also be applied in manufacturing processes to investigate environmental effects. The practical implementation, management, and sustainment of VSM as a lean method can play a critical role in enhancing a manufacturing facility's environmental performance through industrial waste identification, mapping, and elimination [16,22,23].

Compared to Life Cycle Assessment (LCA), the VSM proves to be an improved visualization tool. It can map the complete manufacturing process in a two-dimensional space in a user-friendly way. However, LCA illustrates a limited process view [24,25] that can neglect the seven lean wastes. Uncertainties have also been documented relating to LCA concerning weighted results that confirm the waste management hierarchy and the time-consuming process. The latter requires a highly parameterized model [26], making the VSM a resilient, innovative, lean manufacturing technique and resource-efficient method. The VSM can be used to map industrial waste streams, identify industrial waste inefficiencies, and contribute to reducing, optimizing, improving, and promoting the sustainability of industrial waste management [6,7,13,27].

Therefore, a crucial need exists in the iron and steel industry in developing countries in Southern Africa to identify, apply, and evaluate manufacturing waste as non-value adding activities [28–30]. Considering the vital role of iron and steelmaking in South Africa, the industry remains a critical strategic industry representing 1.5% of its Gross Domestic Product (GDP) and accounts for some 190,000 jobs [31]. The South African steel industry's value chain multiplies the value of iron ore in South Africa by a factor of four. It is further also vital to energy and water-supply infrastructures, amongst others [31]. The VSM method can help the iron and steel industry eliminate waste, achieve zero-waste goals in a lean manufacturing environment, and contribute to the triple bottom line.

The VSM was adopted from the conventional VSM approach [6,27,30,32,33] and applied as a management supporting tool to identify and evaluate industrial waste flow in the iron and steel industry. Mapping of applicable iron and steel waste fractions and material flow generation was adapted. Horizontal performance analysis on specific material efficiency was conducted for each industrial waste segment. The vertical analysis of the waste process and overall efficiency in each subprocess was aligned to the applicable sub-process performance measurements. There is limited research published and proven methods, especially in developing countries in Southern Africa, to promote environmental and operations improvement [6] and incorporate zero waste in a systematic review approach.

The adapted VSM study is considered the first to systematically assess industrial waste flow in the iron and steel industry as a lean manufacturing and zero-waste method [32]. The study's objective was to demonstrate the application of VSM as a supporting management tool to identify and evaluate industrial waste flow in the iron and steel industry at a Southern Africa case study. Following the first year of implementation, waste was reduced by 28%, and waste removal cost by 45%. Implementing the VSM method demonstrated cost savings and reduced waste flow within the study's first year. The initial waste generation reduction target of 5% per annum was exceeded. Demonstrating the application and

effectiveness of VSM as a supporting management tool in the iron and steel industry to identify and evaluate industrial waste flow is a first for Southern Africa. It provides further potential for applying industrial waste reduction in other developing economies and other manufacturing sectors.

## 2. Materials and Methods

The VSM method comprises three main steps. The first step was to collect and verify waste generation and flow data as the VSM data input step. The second step comprises of three phases as the VSM Data Analysis Process, namely, phase 1: Mapping of waste generation and fractions, Phase 2: Horizontal performance analysis—material efficiency for each segment and, Phase 3: Vertical analysis of the waste process and overall efficiency in each subprocess. The third step included the compilation of the actual and future state maps.

### 2.1. Step 1: VSM Data Input Required

Data collection on industrial waste took six years at a selected iron and steel case study facility combined with site-specific waste management system audits. Industrial waste generation data as waste volumes and information generated from analyzed waste samples were used as VSM input data. Laboratory analysis of critical industrial waste streams was analyzed to determine the risk profiling and hazardous nature of the waste generated at the iron and steel facility. The collected data were subjected to a VSM analysis for purposes of mapping waste generation and fractions to determine material efficiency for each waste segment and then to determine the overall efficiencies in each waste sub-process of the iron and steel case study.

### 2.2. Step 2: VSM Data Analysis Process

The collected waste data were subjected to a VSM analysis in three pivotal phases.

#### 2.2.1. Phase 1: Mapping of Waste Generation and Fractions

The phase 1 analysis included mapping waste generation and fractions to analyze sub-processes and processes to visualize improvement potentials in an informal and non-detailed way. The waste management system of the iron and steel facility was divided into selected sub-processes, according to [6], in terms of resources, handling, movements, and inventories. It included both industrial process and general waste. Data were collected on each sub-process and guided by data available on inventories (waste generation data), resources, handling, costing, and movements.

#### 2.2.2. Phase 2: Horizontal Performance Analysis—Material Efficiency for Each Segment

In phase 2, an essential indication of achieving material efficiency is to reduce waste and avoid unnecessary use of raw material in the manufacturing process and waste creation. The waste management system's material efficiencies and activities were examined in phase 2, and the waste activities were identified. To understand the waste material flows and to set Key Performance Indicators (KPIs), the analysis separated the key waste types into main waste segments based on the critical industrial general and process waste streams applicable to the specific site.

The waste efficiency was calculated with the formula (Equation (1)) as a valid approximation [6,33] where the waste efficiency (%) equals either product weight divided by incoming waste weight or product weight divided by the sum of the waste and product weight.

$$\text{Waste efficiency (\%)} = \text{product weight/incoming waste weight} = \text{product weight/(waste weight + product weight)} \tag{1}$$

The performance of each waste segment was monitored separately for each segment so that the potential for improvements can be monitored annually. Each segment's recycling

rate was calculated by dividing the amount of waste recycled by the waste segment total. The average segment treatment cost was calculated by dividing the segment's cost by the waste segment total. The weight per produced unit was calculated by dividing the total waste segment by produced crude steel.

2.2.3. Phase 3: Vertical Analysis of the Waste Process and Overall Efficiency in Each Subprocess

In phase 3, the critical sub-process performance measurements, namely service efficiency, cost efficiency, and overall efficiencies, were analyzed in terms of waste management sub-processes [6] and included bins and collection points, internal handling, external transportation, and treatment. The vertical analysis of the waste process and overall efficiency in each sub-process was applied to industrial general and process waste streams. The vertical analysis results were presented as an averaged monthly performance measurement to understand the average monthly process and overall sub-process efficiencies.

*2.3. Step 3: Compiling the VSM Maps*

The final step in the data analysis included the compilation of the VSM maps. The waste data analyzed was then used to compile an actual and future state map of waste management at the iron and steel facility. The inputs and outputs focused on a facility-wide level and included internal and external disposal, internal waste handling, transportation and treatment, recycling, and final disposal. The actual and future state maps were used to reveal opportunities to reduce costs, improve waste minimization, save time, reduce waste, optimize and identify zero waste opportunities, and improve environmental performance.

## 3. Case Study Application

The case study was based on an iron and steel facility in South Africa. The iron and steel facility was founded in 1957 and is a vertically integrated iron and steel manufacturing facility that produces around 1 million tons of steel blocks annually. The case study was selected because of the availability of actual recorded data.

The VSM, as step 2, was applied in three phases after step 1 was completed.

*3.1. Step 1: VSM Data Generation*

The data collection process for phase 1 consisted of a waste management system and -facilities audit that was based on the following: (1) local regulatory requirements, (2) best practicable environmental options and sustainability requirements, (3) investigating waste generation volumes, (4) waste sampling, (5) laboratory analysis outcomes and (6) waste risk profiling.

*3.2. Step 2: VSM Data Analysis Process*

3.2.1. Phase 1: Mapping of Iron and Steel Waste Fractions and Generation

Following the data collection for phase 1, an analysis was conducted that included mapping waste generation and fractions to analyze sub-processes and processes to visualize improvement potentials in a non-detailed and informal way. The six sub-process division included four sub-processes in material flow and two sub-processes in knowledge flow. The six sub-processes are illustrated in Table 1.

**Table 1.** Waste management sub-processes identified [6] that apply to the iron and steel case study.

| | Sub-Process | Description |
|---|---|---|
| **Material flow** | Workplace waste bins and signs | Data on bins and waste containers and collection layouts. |
| | Internal handling and collection points | Collection layouts and internal and external handling of waste material. |
| | Transport | Internal or external transport by contractors and on-site personnel. |
| | Final treatment | Final treatment and disposal operations were analyzed by type of disposal and location. |

**Table 1.** *Cont.*

| | Sub-Process | Description |
|---|---|---|
| **Information management** | Knowledge flow | Data was collected using stakeholder feedback (as contained in completed Environmental Impact Assessments), on-site observations, historical data records. The improvement process was documented. The waste management process's overall efficiency was calculated to estimate the improvement work and the information system efficiency. |
| | Improvement work | Improvement work was guided by process efficiency data such as general and process waste recycling and re-use. |

Data were collected on each sub-process and guided by data available on inventories (waste generation data), resources, handling, costing, and movements. Sub-process (1) workplace bins and signs were mapped and documented in tables and layouts, including data on the type, size, number of bins or containers, costs associated with bin rental, ownership and maintenance, and inefficiencies associated with waste handling. The collection points and layouts of waste containers and equipment for waste storage, separation, and sorting were also mapped and included maintenance and cost of ownership or renting. In sub-process (2), the internal handling of waste material, from operations internally and externally waste-handling by contractors, was mapped to include data on human resources time and waste movement costs. In sub-process (3), external transportation was mapped by the off-site type and cost for each material segment. Sub-process (4), final treatment operations were analyzed by type of disposal or treatment, cost, and location. For sub-processes (5) and (6), information management data for knowledge flow and improvement work was collected by completed Environmental Impact Assessments (EIAs) and waste licensing documentation, historical data records, and process efficiencies data such as waste recycling and re-use data. The efficiency of knowledge flow and improvement work was estimated based on the processes' overall efficiency.

3.2.2. Phase 2: Horizontal Performance Analysis—Iron and Steel Material Waste Efficiency for Each Waste Segment

The waste management system and activities examined in phase 2 comprised 47 different waste activities. To understand the waste material flows and to set KPIs, the analysis separated the waste types into five main segments in the industrial process and general waste:

- Metals (process waste)
- Hazardous waste (process waste)
- Non-hazardous waste (process waste)
- General waste
- Liquid waste (process waste)

The five segments were chosen based on the industrial operations' waste activities characteristic at the iron and steel facility and different waste materials generated by the facility. The waste efficiency (Equation (1)) was calculated with a formula as a valid approximation [6,33], where material efficiency (%) equals either product weight divided by incoming material weight or product weight divided by the sum of the waste and product weight.

Waste efficiency (%) = product weight/incoming material weight = product weight/(waste weight + product weight)

The performance of each waste segment was analyzed so that improvements can be monitored over time.

The need to monitor and measure actual iron and steel waste generation and services was essential to control and facilitate iron and steel waste operations efficiently. According to [6] each waste segment's performance should be monitored separately for each segment so that the potential for improvements can be monitored over time. The recycling rate in each segment and cost per segment also needs to be monitored [6,33]. Each segment's recycling rate was calculated by dividing the amount of waste recycled by the waste segment total. The average segment treatment cost was calculated by dividing the segment's cost by the waste segment total. The weight per produced unit was calculated by dividing the waste segment total by produced crude steel. Hence, the inclusion of iron and steel waste segment performance measurements as detailed in Table 2.

**Table 2.** Iron and steel waste segment performance measurements [6] adapted.

| Iron and Steel Segment Indices | Calculation | Unit |
|---|---|---|
| Weight (W) sorting rate | $W^1_{(sorted)} / W_{(segment\ total)}$ | % |
| Weight (W) per produced unit (P) | $W_{(segment\ total)} / P^2$ | Ton/# |
| Average segment treatment cost (C) | $C^3_{(segment\ total)} / W_{(segment\ total)}$ | SEK/ton |

1—Weight, 2—Cost, 3—Produced unit.

### 3.2.3. Phase 3: Vertical Analysis of the Iron and Steel Waste Process and Overall Waste Efficiency in Each Iron and Steel Waste Sub-Process

In phase 3, the critical sub-process performance measurements, namely service efficiency, cost efficiency, and overall efficiencies, were analyzed in terms of waste management sub-processes, namely bins and collection points, internal handling, external transportation, and treatment. The vertical analysis of the waste process and overall efficiency in each sub-process were applied to both general and process waste streams. The vertical analysis results were presented as an averaged monthly performance measurement to understand the average monthly process annually and overall sub-process efficiencies.

The waste service efficiency as a critical sub-process performance measurement was measured by four waste management sub-processes characteristic of the iron and steel facility's critical sub-processes. The number of containers, including collection points, were divided by the amount of waste, in tonnage, in bins to obtain the number of bins available per tonnage of waste generated. Person-hours determining service efficiency, related to internal handling, were divided by the amount of waste (tonnage) to obtain the number of person-hours required to service a tonnage of waste. Service efficiency in terms of waste transportation was determined by dividing the number of trucks per waste collected by the amount of waste to be transported from collection points. The final treatment's waste service efficiency was determined by dividing the amount of waste recycled by the amount of waste generated.

As a critical sub-process performance measurement, the cost-efficiency was measured by three waste management sub-processes (Table 3). The cost-efficiency in terms of containers was determined by dividing the costs (maintenance and rental) of waste containers by collecting waste. As a waste management sub-process of cost efficiency, internal handling and transportation of waste were determined by dividing the costs associated with person-hours (personnel required to manage waste) by the amount of waste to be managed. Cost efficiency in terms of external handling and transportation of waste was determined by dividing the waste transport cost by the amount of waste transported. The cost-efficiency of treatment was calculated by dividing the cost of waste treatment by the total amount of waste.

**Table 3.** Iron and steel waste sub-process performance measurements [6].

| | Bins | Internal Handling | Collection Points | External Transportation | Treatment |
|---|---|---|---|---|---|
| **Service efficiency** | $\#_{(bins)}/W$ [1] (waste in bins) | Person-h/W | $\#_{(containers)}/W$ (waste in containers) | $\#_{(trucks)}/W$ (waste transported) | $W_{(recycled)}/W_{(sum)}$ and $W_{(landfilled)}/W$ (generated) |
| **Cost efficiency** | $C^{[2]}_{(bins)}/W$ (waste in bins) | $C_{(person-h)}/W$ | $C_{(equipment)}/W$ (waste in equipment) | $C_{(transports)}/W$ (waste transported) | $C_{(treatment)}/W_{(sum)}$ |
| **Overall effectiveness** | $C_{(bins)}/P$ [3] | $C_{(person-h)}/P$ | $C_{(equipment)}/P$ | $C_{(trucks)}/W$ (waste transported) | $C_{(treatment)}/P$ |

1—Weight, 2—Cost, 3—Produced unit.

Three waste management sub-processes measured the overall effectiveness as a critical sub-process performance measurement. The containers' overall effectiveness (as waste infrastructure) was determined by dividing the costs associated with the waste containers by the manufactured weight of produced crude steel. The overall effectiveness of the internal handling of waste was determined by dividing the cost of personnel responsible for waste management by the manufactured weight of produced crude steel. External transportation's overall effectiveness was determined by dividing the cost associated with waste transportation by the weight of waste transported. The overall effectiveness of treatment was determined by dividing the cost of waste treatment by the manufactured weight of produced crude steel.

The waste service evaluations reflect the effectiveness and quality of the waste service provided. The sub-process measurements were subordinated to the overall performance measures of process and general waste separately to avoid sub-optimization and confusion between the iron and steel facility's two main waste categories. By evaluating the average treatment cost and sorting degree in each waste segment, best practices and gaps could be identified. The applicable waste management sub-processes for general waste included containers, internal handling, external transportation, and treatment. The applicable waste management sub-processes for process waste were limited to on-site disposal facilities, internal handling, and internal treatment as process waste is managed, treated, and disposed of on the site of the iron and steel facility. Limited use of external contractors transporting and treating waste off-site applies to the handling and treatment of process waste.

By evaluating the average treatment cost and sorting degree in each iron and steel waste segment, best practices and gaps could be identified on iron and steel waste segments (Kurdve et al. 2017). To contribute to an iron and steel facility's lean practices, the focus was placed on the handling and using non-value adding (NVA) and non-productive output (NPO) materials. When improvement work is conducted on iron and steel waste management, the various inefficiencies can be addressed simultaneously. Initially, the overall efficiency needs to be analyzed and determined, and after that, the efficiency of each sub-process. Performance measurements are illustrated in Table 3 for each of the iron and steel waste management sub-processes. The performance measurements had to reflect the quality and effectiveness of services applied. In order to prevent suboptimization, the sub-processes measurements were subordinated to the overall performance measurements. An example to demonstrate the suboptimization prevention is if only one bin is used, which contains all types of iron and waste, the bins efficiency measure is acceptable. However, sorting and final treatment costs can give a non-optimal result that includes internal transportation [6].

## 4. Results

### 4.1. Phase 1: Mapping of Waste Generation and Fractions

Per annum, an approximate amount of 314 tons of iron and steel general compactable waste was generated in the case study that primarily demonstrates similar composition to general municipal waste. Table 4 illustrates the leading waste management system components, including crude steel production, for analyzing this section. The amount of iron and steel waste generated per ton of crude carbon steel as 2.15 tons, was higher than the world steel average of 0.3 tons per one ton of crude carbon steel and was principally related to inefficient iron and steel waste disposal and re-use practices, process inefficiencies, challenges experienced in the waste management and operational systems and aging manufacturing infrastructure.

**Table 4.** The horizontal performance analysis—iron and steel waste efficiency for general waste at the case study over six years.

| Proposed Segment Indices—Iron and Steel General Waste (Per Annum) | Calculation | Unit |
|---|---|---|
| Iron and Steel Waste Recycling Rate | W (Recycled)/W (Segment Total) | % |
| | W (349)/W (315 + 558) | 40% |
| | W (349)/W (873) | |
| Weight Per Produced Unit | W (Segment Total)/P | ton/# |
| | W (873 t)/P (686,632 t) | 0.001 t/Pt |
| Average Iron and Steel Waste Segment Treatment Cost | C (Segment Total)/W (Segment Total) | SEK/ton |
| | C (USD 180,279)/W (873 t) | USD 206/t waste |

Iron and steel process waste recycling in the case study amounted to 27% where steel slag is screened and primarily re-used for application as agricultural lime in the agriculture industry. The slag is mechanically treated through screening activities on the iron and steel facility site, and the screened material is subsequently stockpiled. Iron slag recycling amounts to 23%, where recycled iron is mainly recovered and utilized in the ironmaking process at the ironmaking units. The screening is done on-site as mechanical treatment. Iron slag recycling for titanium recovery amounts to only 20%. The recycling percentages for both iron and steel slag can be over 60%, once local markets have been secured for aggregate use in road construction, cement manufacturing, brick making, as an addition in construction materials, and further application and use in the agriculture sector [34–46].

Figures 1 and 2 respectively indicate the waste generation and recycling rates of iron and steel waste, indicating a considerable opportunity to increase recycling rates in particular iron and steel process waste streams.

### 4.2. Phase 2: Horizontal Performance Analysis—Waste Efficiency for Each Segment

The iron and steel facility's horizontal performance measurement analysis regarding the recycling rate and cost of the iron and steel waste fractions (efficiency indicators) for each iron and steel waste segment (general and process waste) are described in Tables 4 and 5.

The horizontal performance analysis conducted at the case study for iron and steel waste stream segments was used to identify potential improvements for the key waste segments in the complete iron and steel waste management process. The horizontal performance analysis (Tables 4 and 5) at the iron and steel case study facility indicated substantive low recycling rates for the general waste stream at 40% and a 22% recycling rate for the process waste stream. The low recycling rates can be ascribed to several factors such as (1) waste stream separation inefficiencies, (2) recycling contractor unreliability, and (3) national waste legislation challenges that complicate and restricts downstream use of iron and steel

process waste. The weight per produced unit also indicates local efficiency challenges where, when compared to the global average of the World Steel Organization at 0.3 t per produced unit, the iron and steel facility case study exceeds two tons per produced unit (that includes the general waste at 0.001 t/Pt).

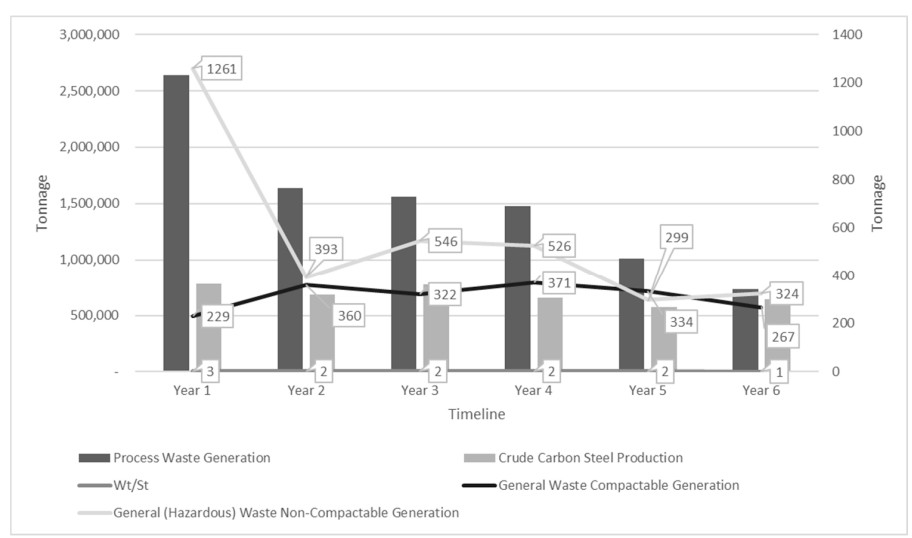

**Figure 1.** Iron and steel waste generation vs. crude steel production (annual averages over six years).

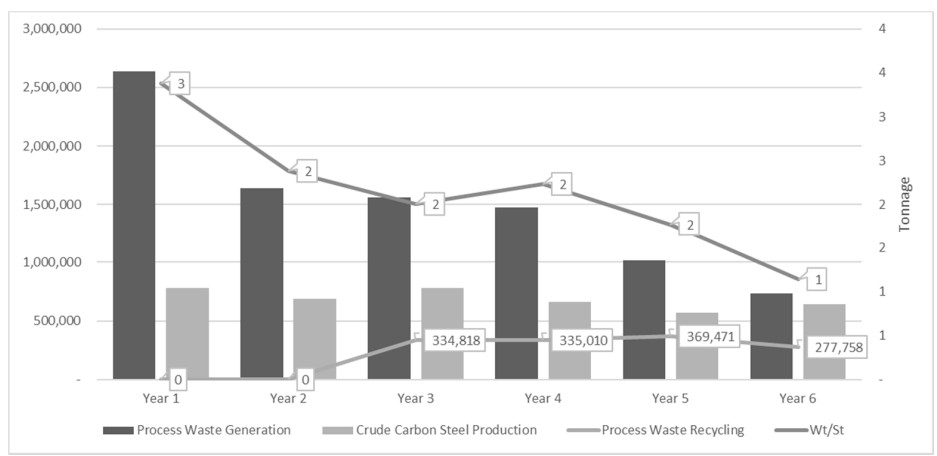

**Figure 2.** Iron and steel process waste generation and recycling volumes per annum.

**Table 5.** The horizontal performance analysis—iron and steel waste efficiency for process waste at the case study over six years.

| Proposed Segment Indices—Iron and Steel Process Waste (Per Annum) | Calculation | Unit |
|---|---|---|
| Iron and Steel Waste Recycling Rate | W (Recycled)/W (Segment Total) | % |
| | W (329,264)/W (1,510,064) | 22% |
| Weight Per Produced Unit | W (Segment Total)/P | ton/# |
| | W (1,510,064)/P (686,632 t) | 2 t/Pt |
| Average Iron and Steel Waste Segment Treatment Cost | C (Segment Total)/W (Segment Total) | SEK/ton |
| | C (USD 891,958)/W (1,510,064 t) | USD 0.6/t waste |

Optimally, compared with the current best practice or the global average of 0.3 t/Pt, the total iron and steel waste generation per annum should be approximately 205,990 tons. The actual iron and steel waste management and treatment costs (Tables 4 and 5) for the general waste were USD 206/ton, including iron and steel off-site waste handling and waste disposal at the case study facility. The average treatment cost that includes on-site handling and disposal for industrial process waste was USD 0.6/t. The results indicate the associated costs of off-site iron and steel waste handling and external waste treatment. It can be concluded from the results that the horizontal performance analysis indicates principally that on-site iron and steel waste disposal was taking place at the case study facility with limited focus on zero waste.

### 4.3. Phase 3: Vertical Analysis of the Waste Process and Overall Efficiency in Each Sub-Process

The completed VSM vertical analysis results explicitly conducted in the case study facility are illustrated in Tables 6 and 7. An actual and future state VSM can be compiled following the completion of the vertical analysis of the sub-process waste process and overall efficiency.

**Table 6.** VSM results for general waste in the case study averaged over six years.

| Subprocess Performance Actual Measurements (Per Month) | Bins | | Internal Handling | | External Transportation | | External Treatment | |
|---|---|---|---|---|---|---|---|---|
| **Service Efficiency** | # (bins)/W (waste in bins) | 60 bins/110 tons | Person-h/W | 880/110 tons | # (trucks)/W (waste transported) | 5/110 | W (recycled)/W (sum)(sum) | 349 tons/872 tons |
| | 0.55 bins available per ton of waste generated | | 8 person-hours available to manage one ton of waste generated | | 0.05 trucks available per one ton of waste generated | | For every ton of waste generated, 0.4 tons are recycled | |
| **Cost Efficiency (Unit of Cost expressed in USD)** | C (bins)/W (waste in bins) | 11,905/110 tons | C (Person)/W | 14,881/110 tons | C (transport+disposal)/W (waste transported) | 15,739/110 tons | C (treatment-disposal & transport)/W (sum) | 15,739/110 tons |
| | To maintain the bins per one ton of waste generated it costs USD 108 | | Labor costs amount to USD 135 per one ton of waste generated | | To transport and treat one ton of waste, the cost is USD 143 | | To transport and treat one ton of waste amounts to USD 143 | |
| **Overall Effectiveness (Unit of Cost expressed in USD)** | C (bins)/P | 11,905/57,219 tons | C (Person-h)/P | 14,881/57,219 | C (trucks)/W (waste transported) | 15,739/110 tons | C (treatment)/P | 15,739/57,219 |
| | It costs USD 0.21 to maintain the bins per each one ton of crude steel produced | | The labor costs are USD 0.26 per ton of crude steel produced | | The treatment and transport costs per one ton of waste generated amount to USD 143 | | For each ton of crude steel produced, it costs USD 0.28 to transport and finally treat one ton of waste | |

**Table 7.** VSM analysis results for process waste in the case study averaged over six years.

| Subprocess Performance Actual Measurements (Per Month) | Disposal Facilities | | Internal Handling | | Internal Treatment | |
|---|---|---|---|---|---|---|
| **Service Efficiency** | # (disposal facilities)/W (waste generated) | 46 facilities /125,839 tons | Person-h/W | 720/125,839 | W (recycled)/W (sum)(sum) | 27,439 tons/125,839 tons |
| | 0.0004 facilities are available per one ton of waste generated | | For each ton of waste generated, 0.006 person-hours are available to deal with such waste | | For each one ton of waste generated, 0.22 tons of waste is recycled or re-used | |
| **Cost Efficiency (Unit of Cost expressed in USD)** | C (disposal facilities)/W (waste generated) | 77,958/125,839 tons | C (person)/W | 14,881/125,839 tons | C (treatment-disposal & transport)/W (sum) | 77,958/125,839 tons |
| | The cost associated with managing one ton of waste is approximately USD 0.62 | | To manage (labor costs) one ton of waste costs USD 0.12 | | The cost associated with managing one ton of waste is approximately USD 0.62 | |
| **Overall Effectiveness (Unit of Cost expressed in USD)** | C (disposal facilities)/P | 77,958/57,219 tons | C (person-h)/P | 14,881/57,219 | C (treatment)/P | 77,958/57,219 |
| | For each one ton of crude steel produced, it costs USD 1.36 to manage and dispose of waste | | Labor costs amount to USD 0.26 per each one ton of crude steel produced | | For each one ton of crude steel produced, it costs USD 1.36 to manage and dispose of waste | |

The results indicate for process waste, for each ton of waste generated, only 0.4 tons are recycled compared to industrial process waste, where only 0.22 tons are recycled per ton of waste generated. The recycling rate is consequently 40% for general waste and only 22% for process waste. For each ton of crude steel produced, the transport costs amount to USD 0.28 to transport and finally treat one ton of general waste with an average treatment cost of USD 143 per ton of waste. The cost associated with managing one ton of process waste on site amounts to approximately USD 0.62. The difference in costs is related to on and off-site treatment facilities and external disposal costs. Further, for each ton of crude steel produced, it costs USD 1.36 to manage and dispose of process waste.

As part of the vertical analysis of iron and steel waste sub-process efficiency and overall best-practice analysis, potential improvements were identified for the key iron and steel waste segments and presented in the form of an existing and future-state VSM. As part of the vertical analysis, the iron and steel facility was evaluated by investigating the waste recycling rates, average waste treatment costs, service efficiencies, and overall effectiveness. Potential iron and steel waste management process improvements were identified in all five iron and steel waste sub-processes and are indicated in the future-state VSM. The actual and future state of iron and steel waste VSM is indicated in Figures 3 and 4.

*4.4. Step 3: VSM Maps*

4.4.1. Actual State VSM Map

Step one and two of the VSM data input and analysis were used to compile the actual state map (Figure 3). Activities associated with sub-process performance were included on the map to display waste flow and industrial waste generation costs visually. An immediate opportunity could be identified by generating an actual state map to reduce waste generation and costs associated with waste activities. After the first year of applying the VSM method, industrial waste was reduced by 28%, and costs associated with industrial waste removal was reduced by 45%. Externalizing the iron and steel facility's waste operations was found to be cost-intensive. By utilizing existing personnel, waste costs

could be reduced within a year with the added advantage of having full control over the flow and management of waste. Visualizing and analyzing key waste streams, recycling status, waste flows, and investigating waste generation areas helped identify double handling areas with unnecessary associated costs. Other immediate areas of improvement included using waste infrastructure strategically to minimize waste mixing and optimize waste transport logistics.

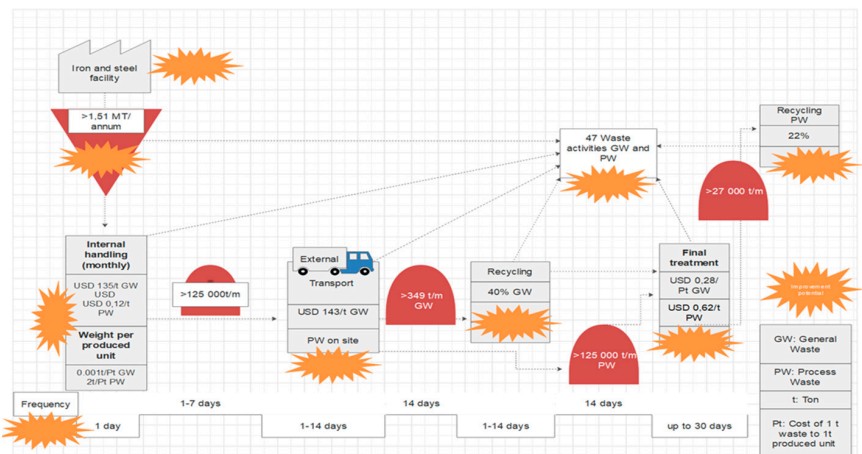

**Figure 3.** Iron and steel VSM: Actual State in year one at the case study.

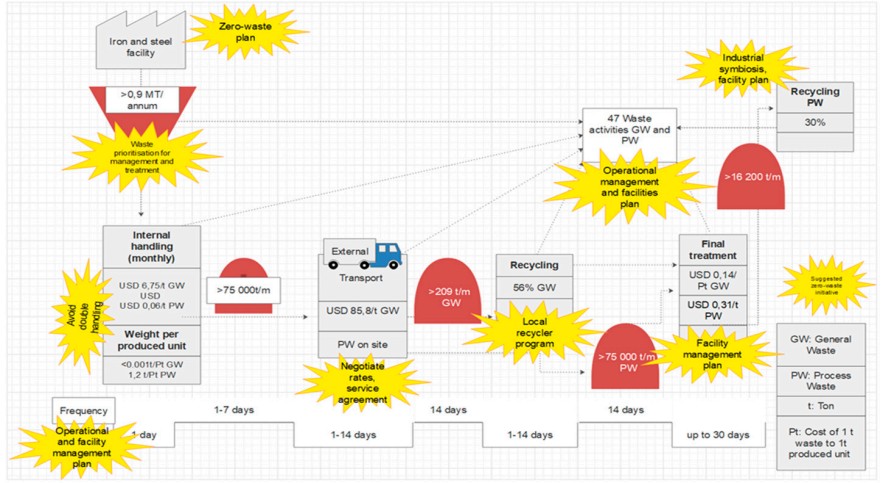

**Figure 4.** Iron and steel VSM: recommended Future State in the case study.

The following iron and steel process waste management system components have been identified through compiling the actual state map. The areas of improvement require an increase in the efficiency of the iron and steel waste management and treatment system at the case study site:

(1) Investigating and improving the company culture;
(2) Recording, monitoring, and recording the cost of iron and steel process waste;
(3) Accountability of the various plant divisions being accountable general and process waste generated;
(4) Compliance with regulatory requirements;
(5) Development and maintenance of an integrated waste data system;
(6) Iron and steel waste infrastructure planning, implementation, auditing, monitoring, and maintenance;
(7) Iron and steel waste streams characterization, monitoring, and management;
(8) Contaminant management of on-site waste facilities and mitigating environmental externalities;

(9)    Iron and steel process waste treatment options and methods;

(10)   Phased and prioritization of site remediation of contaminated sites;

(11)   Implementing integrated management practices associated with general waste;

(12)   Implementing sustainability practices and zero waste initiatives.

### 4.4.2. Future State VSM Map

A future state map (Figure 4) was also compiled to visualize the desired state of industrial waste generation at the iron and steel facility in the case study area. The future state map's timeframe was set as five years and included zero waste initiatives that should be considered for implementation. The target for industrial waste generation reduction was set at 40%, and the target for reducing costs associated with waste generation was set at 50%. The targets were developed based on the actual state VSM map and the successes achieved in industrial waste generation reduction and cost reduction in the first year of implementation. Proposed zero-waste initiatives include developing an industrial facility-level zero-waste plan adopted by the iron and steel facility's highest authority. It will be crucial that divisional adoption and accountability is set for achieving the objectives of the strategy. An operational facility and management strategy need to form part of the zero-waste plan to guide the operations to design, implement, monitor and evaluate divisional waste management programs. It is also suggested that each division in an iron and steel facility compile their own actual and future state map that forms part of the facility level actual and future state map. In that way, opportunities and challenges can be identified on a divisional level that can positively impact the facility's zero-waste goals.

### 5. Conclusions

When iron and steel waste is generated in developing countries, it indicates a level of process inefficiencies, inadequate management processes, and limited sustainability practices. To successfully manage any business, resources and time need to be adequately managed, and waste needs to be reduced. However, when industries visualize industrial waste flows, sustainable and efficient resource flow management can be implemented, and zero-waste initiatives initiated. Following the first year of implementation of the VSM, waste was reduced by 28%, and waste removal cost by 45%. Implementing the VSM method demonstrated cost savings and reduced waste flow within the study's first year. The initial waste generation reduction target of 5% per annum was exceeded. Applying a VSM in the iron and steel industry does not only improve the VSM visibility. However, a VSM can also improve the iron and steel industry's sustainability performance by providing transparent sustainability information to stakeholders through visualized actual and future state maps. A VSM can promote cost-savings and the implementation of zero waste initiatives, zero-waste systems performance can be monitored, opportunities can be identified to avoid and minimize waste through implementing operational and facility management plans, and continuous improvement of lean production activities can be promoted through optimizing operational activities to reduce waste. Recycling opportunities can be identified, such as including communities in local entrepreneurial initiatives. The outcome of the VSM can be subsequently applied in developing decision support programs in industrial waste management in iron and steel facilities in developing countries to prioritize industrial waste management and treatment system components to promote a zero-waste footprint.

The VSM method as a management tool's positive aspects includes visually demonstrating and interpreting industrial waste flows at all levels in the facility, including economic indicators and graphical indicators representing waste movements and other activities. The actual and future maps generated can be used in awareness and capacity building programs where local communities are empowered and supported to participate in waste recycling programs. The VSM can be applied in each division to generate their own actual and future state maps to contribute to the facility level zero-waste goals and also to support lean manufacturing programs. The case study's results are limited as it was only applied to one case study, and additional studies should be done by applying the

VSM to other case studies in the manufacturing sector. Another limitation of this study included the inclusion of different production flows, complete facility layout representation, and specifically queues and movements due to layout. However, for the purpose of this study, it was excluded. Future research should consider applying the VSM to different manufacturing and industrial facilities, focusing on achieving zero-waste and to promote the movement towards the circular economy. Future research should be considered to integrate the VSM in systems engineered methods to enable integrated decision-support for waste management and treatment systems in industrial systems.

**Author Contributions:** Conceptualization, Y.S.; methodology, Y.S.; validation, Y.S.; investigation, Y.S.; resources, Y.S.; data curation, Y.S.; writing—original draft preparation, Y.S.; writing—review and editing, Y.S. and P.O.; visualization, Y.S. and P.O.; supervision, P.O. and V.S.; project administration, Y.S. and P.O.; funding acquisition, Y.S. All authors have read and agreed to the published version of the manuscript.

**Funding:** This research was funded by the University of the Free State and the Baoberry Centre of Innovation.

**Institutional Review Board Statement:** Not applicable.

**Informed Consent Statement:** Not applicable.

**Data Availability Statement:** MDPI Research Data Policies.

**Acknowledgments:** The authors would like to thank Marthie Kemp for her administrative support and also Lucas Mlangeni and Anre Weststrate for their support in this study.

**Conflicts of Interest:** The authors declare no conflict of interest.

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
