# Peer review of "Value Stream Mapping as a Supporting Management Tool to Identify the Flow of Industrial Waste: A Case Study"

_sustainability, doi:10.3390/su13010091_

Round 1

Reviewer 1 Report

It´s an interesting wiev about what happnes in many sectors where they coulb improuve the eficiency with not a high effort and in a cheap way. In them it could be completed previously with ther LEAN manufacturing techniques

Author Response

Dear Reviewer 1,

Thank you for the time to review our manuscript. 

Response to reviewer 1 comments:

Point 1: It´s an interesting wiev about what happnes in many sectors where they coulb improuve the eficiency with not a high effort and in a cheap way.

Response 1: Thank you. I agree.

Point 2: In them it could be completed previously with ther LEAN manufacturing techniques.

Response 1: I also agree and it is a very good point. I have included this in the reviewed manuscript in line 442.

Thank you again sincerely.

Regards,

Yolandi Schoeman

Reviewer 2 Report

The paper describe the result of application tools for sustainable handling and management of waste in this case industrial waste form iron and steel processing. The paper brings interesting information especially for developing counters but not only.  Nice information about benefits of Value Stream Mapping implementation are given and more over some important recommendation for future are presented.

Paper should be published after same amendments:

  1. English correction
  2. Objective of the paper are missing – should be formulated
  3. Conclusions should be rewritten in the forma to answer for the objective and give clear benefits of VSM application

Author Response

Dear reviewer 2,

Thank you for taking the time to review the paper. 

Herewith a response to your comments.

Point 1 : English correction

Response: The full paper was checked again for english correction and adapted.

Point 2: Objective of the paper are missing - should be formulated

Response: The objective was made clearer and rephrased in line 92 to 94 as "The study's objective was to demonstrate the application of  VSM as a supporting management tool to identify and evaluate industrial waste flow in the iron and steel industry at a Southern Africa case study." 

Point 3: Conclusions should be rewritten in the forma to answer for the objective and give clear benefits of VSM application

Response: The answer for the objective and the clear benefits of the VSM application is included in line 427 to 446 as: 

Applying a VSM in the iron and steel industry does not only improve the VSM visibility. However, a VSM can also improve the iron and steel industry's sustainability performance by providing transparent sustainability information to stakeholders through visualized actual and future state maps.  A VSM can promote cost-savings and the implementation of zero waste initiatives, zero-waste systems performance can be monitored, opportunities can be identified to avoid and minimize waste through implementing operational and facility management plans, and continuous improvement of lean production activities can be promoted through optimizing operational activities to reduce waste.  Recycling opportunities can be identified, such as including communities in local entrepreneurial initiatives. The outcome of the VSM can be subsequently applied in developing decision support programs in industrial waste management in iron and steel facilities in developing countries to prioritize industrial waste management and treatment system components to promote a zero-waste footprint.

The VSM method as a management tool's positive aspects includes visually demonstrating and interpreting industrial waste flows at all levels in the facility, including economic indicators and graphical indicators representing waste movements and other activities. The actual and future maps generated can be used in awareness and capacity building programs where local communities are empowered and supported to participate in waste recycling programs.  The VSM can be applied in each division to generate their own actual and future state maps to contribute to the facility level zero-waste goals and also to support lean manufacturing programs.

However the following was added to the conclusion " Following the first year of implementation of the VSM, waste was reduced by 28%, and waste removal cost by 45%. Implementing the VSM method demonstrated cost savings and reduced waste flow within the study's first year. The initial waste generation reduction target of 5% per annum was exceeded." as line 427-430.

Thank you again for your helpful comments and kind review.

Have a blessed festive season.

Yolandi

Reviewer 3 Report

I would have appreciated more talk on the limitations of the paper. Certainly, one case study is a limitation, even if the results obtained are excellent in that specific case.

Author Response

Dear Reviewer 3,

Thank you very much for taking the time to review our manuscript. It is much appreciated in these challenging times. Thank you also very much for your kind comments.

Point 1: I would have appreciated more talk on the limitations of the paper. Certainly, one case study is a limitation, even if the results obtained are excellent in that specific case.

Response: I have made changes to line 449 to 455 to include a change based on your comments: "This case study's results are limited as it was only applied to one case study, and additional studies should be done by applying the VSM to other case studies in the manufacturing sector. Another limitation of this study included the inclusion of different production flows, complete facility layout representation, and specifically queues and movements due to layout. However, for the purpose of this study, it was excluded. Future research should consider applying the VSM to different manufacturing and industrial facilities, focusing on achieving zero-waste and to promote the movement towards the circular economy. "

Thank you again for your comments and for being part of our journey.

I wish you a blessed festive season.

Kind regards,

Yolandi Schoeman